# The Impact of Medical and Health Fiscal Expenditures on Pharmaceutical Industry Stock Index in China

**DOI:** 10.3390/ijerph191811730

**Published:** 2022-09-17

**Authors:** Chiwei Su, Yiru Liu, Chang Liu, Ran Tao

**Affiliations:** 1School of Business, Wuhan University of Technology, Wuhan 430062, China; 2School of Economics, Qingdao University, Qingdao 266071, China; 3Qingdao Municipal Center for Disease Control & Preventation, Qingdao 266032, China

**Keywords:** medical and health fiscal expenditure, pharmaceutical industry, causal relationship, time-varying

## Abstract

This paper investigates the relationship between fiscal expenditure on health care (*FE*) and the pharmaceutical industry stock index (*SP*) by using full-sample and sub-sample rolling-window bootstrap causality tests. It can be observed that there is both a positive and negative relationship between *FE* and *SP*. *FE* will promote the rise of the pharmaceutical stock market, which proves the Keynesian theory, while the result that *FE* negatively affects *SP* supports the classical theory. In turn, *SP* positively impacts FE, which indicates that the development of the pharmaceutical industry and the increase in medical and health expenditures can promote each other. In addition, the negative influence of *SP* on *FE* suggests that the impact of the pharmaceutical index on fiscal expenditure needs to be judged in conjunction with other events and market conditions. In complex economic conditions, investors can rationally consider the industry situation of the pharmaceutical market and benefit by optimising their investment portfolios. The government can regulate and guide the pharmaceutical industry by adjusting the fiscal expenditure on health care, thereby promoting the sustainable and stable development of the financial market.

## 1. Introduction

Health care is an essential part of people’s livelihood. It is inseparable from the health and life of the people and has always been the focus of the country and the people. In the context of ageing and new urbanisation, the development of China’s medical and health sector should also continue to progress and improve [1]. Given the solid externalities and risks in the medical and health field, the government has a more outstanding obligation to provide primary medical and health care services than simply relying on market forces for adjustment, which means that the government must play a leading role [2]. In other words, the health care industry is largely influenced by government policies. Political and economic shocks, economic uncertainty, increased health care costs, drug innovation, pricing policies, R&D, revised regulations, and market competition all impact the pharmaceutical industry [3,4]. Hao et al. [5] proposed that fiscal policies and reforms significantly affect health spending in China. Gerring et al. [6] showed that disease outbreaks and other public health emergencies can impact the efficiency of fiscal health spending. According to Heidari et al. [1], the pharmaceutical industry is considered to be the most competitive and profitable key industry in the world, and its stock index is favoured by the majority of investors. Hence, investors prefer to gain benefits by understanding the factors affecting the pharmaceutical industry. It is necessary to pay attention to the ways in which government policies affect the pharmaceutical industry. Furthermore, the “healthy China” strategy put forward at the 19th National Congress of the Communist Party of China (CPC) and the policy for ensuring essential people’s livelihood in 2020 demonstrate that the government attaches great importance to people’s health. Therefore, the pharmaceutical industry in China has good development prospects and has attracted much attention from investors. As a tool used by the government to regulate the economy, fiscal policy will influence the Chinese stock market [7]. Nevertheless, there is no clear conclusion as to whether China’s fiscal medical and health expenditure will affect the development of pharmaceutical stocks, whether it has an impact, and whether the influence of different stages on pharmaceutical stocks is the same. Therefore, it is vital to detect the influence of fiscal expenditures (*FE*) on pharmaceutical stocks (*SP*) in different periods, which is very important for investors to construct a reasonable investment portfolio.

The securities market of China has gradually developed over the past two decades. It has been a significant industry in the national economy, extensively promoting the economy’s growth. However, the low entry threshold for companies to go public and the low overall investment quality of traders, mostly retail investors, lead to serious speculation in my country’s stock market. Hence, government intervention in the market is inevitable [7]. Moreover, China’s stock market is immature, and stock prices are prone to ups and downs. The factors affecting stock prices are more complicated than those in developed countries. Therefore, studying the impact of government investment in health care on the stock index of the pharmaceutical industry will help investors to clarify the investment direction. Jin and Qian [8] pointed out that China’s health care expenditure level is relatively low compared with developed countries. The improvement of economic status will promote increasing spending on health care [9,10]. Since the new medical reform in 2009, the scale of Chinese health care expenditure has gradually expanded. From CNY 198.9 billion in health expenditure in 2007 to CNY 20,894 in 2021, China’s medical and health expenditure has increased by about 10.5 times, hence, the increase in public health spending may be a key factor in promoting the development of the pharmaceutical industry. Based on this background, this paper takes the pharmaceutical industry as an entry point, conducts an empirical analysis of Chinese medical and health fiscal expenditure and the pharmaceutical industry stock price index, studies the interaction between the two variables, and the regulation and promotion of fiscal expenditure on the stock market, in order to help improve China’s capital market and improve the real economy. The Chinese pharmaceutical market and companies are increasingly important to the global pharmaceutical industry [11]. Studying the impact of government financial expenditure on the development of the pharmaceutical industry is conducive to the sustainable and healthy development of China’s pharmaceutical industry and enhances the status of China’s pharmaceutical industry in the world.

The stabiliser role of fiscal expenditure and the mechanism of discretionary power can promote the stability of social resources, guide the flow of fiscal funds to public areas such as education, medical care, and environmental protection, and drive economic development [12]. Nijkamp and Poot [13] clarified that increased fiscal spending is beneficial to future economic development. Kotlebova et al. [14] illustrated that fiscal policy can boost the real economy and positively impact the stock market. However, according to Foresti and Napolitano [15], macroeconomic scenarios can also affect stock market responses to fiscal policy. During the financial crisis, the stock index will rise with the increase in fiscal investment, but in other periods, the loose fiscal policy will cause the stock to plummet [15]. For other purposes, such as achieving environmental goals, the government will adopt aggressive fiscal subsidy policies, which can also have an impact on an industry [14]. Afonso and Sousa [16] investigated that government expenditures negatively affect stock markets in the U.S., U.K. and Germany. Chatziantoniou et al. [17] showed that aggressive government spending sends the U.K. stocks lower, but fiscal spending indirectly affects stocks in Germany and the U.S. Jakova [18] illustrated that in Slovakia, Romania, the Czech Republic and Bulgaria, rising stocks lead to lower government spending, in Poland, there is no relationship between stock market performance and government spending, and Hungary’s stock market performance was positively affected by higher public spending. Therefore, the relationship between fiscal spending and the stock market is different in different countries. Ardagna [19] found that the trend of a country’s past fiscal policy and its position in its policy and fiscal policy changes can lead to sharp declines in company stock prices. According to Afonso and Fernandes [20], the changes in financial expenditure will negatively affect the stock price of listed companies. Furthermore, Culter et al. [21] pointed out that public health expenditure reflects the government’s health care policy. That is to say, medical and health policies could lead to an increase or decrease in health care fiscal expenditures. In addition, health care expenditure can affect health [21], while health can contribute to a country or region’s economic growth and development [22,23,24]. The stock market will rise with the development of the economy [25,26]. Therefore, medical fiscal expenditure may affect pharmaceutical industry stocks. However, Chatziantoniou et al. [17] concluded that monetary policy and fiscal policy need to act together to impact the stock market significantly. Ricardo denies the impact of fiscal policy on the stock market. Therefore, the impact of fiscal expenditures on stock prices is uncertain. Furthermore, Shafiullah et al. [27] indicated that stock market declines change government policies in developed and high-income countries, implying that changes in stock markets affect fiscal spending. Therefore, the influence from fiscal expenditure on stock prices is complex, and the impact on the economy is not the same in different periods. So far, there is no unified conclusion. In addition, the changing trend of medical financial expenditure and the stock price index of the pharmaceutical industry is not very obvious, so the relationship between the two variables must be verified through empirical analysis.

This paper has several marginal contributions. First, investors can analyse the stock price trend of the pharmaceutical industry through the state’s financial expenditure on health care. The government can better maintain the stability and development of the pharmaceutical industry through fiscal expenditure. Secondly, this paper takes China as an example to investigate how fiscal expenditures, as a means of macroeconomic regulation and control, affect the stock market trend to verify whether China’s stock market can be used as a “barometer” of economic development. Third, previous studies have only shown the non-constant relationship between fiscal expenditures and stock trends or economic development while ignoring the impact of different categories of fiscal expenditures on specific industries and ignoring the nature of time-varying model parameters. Therefore, to bridge the time-varying interaction gap between *FE* and SE, the bootstrap sub-sample rolling window will be employed for estimation.

The rest of this study is organised as follows. Section 2 and Section 3 present the literature review and theoretical basis. Section 4 discusses the method used in this paper. Section 5 and Section 6 deal with data and empirical results. Section 6 deals with the results and key findings of the empirical analysis. The last section concludes the paper.

## 2. Literature Review

Laopodis and Sawhney [28] found a multi-month negative correlation between the U.S. fiscal deficit and stock price volatility. Darrtat [29] used Canadian data from 1960 to 1984 to conduct an empirical analysis. The results show that the stock returns of listed companies are greatly affected by changes in fiscal policy. Fiscal policy can have a noticeable negative impact on the stock prices of listed companies. Tavares and Valkanov [30] considered that in the U.S. market, some of the unexpected returns in stock returns were caused by changes in fiscal policy. Ardagna [19] analysed the data on stock price fluctuations and interest rate changes when the fiscal policy changed significantly from 1960 to 2002 in OECD countries. The tendency of a country’s past fiscal policy and its position in the country’s policy, and changes in fiscal policy will lead to a substantial drop in interest rates and company stock prices, such as large-scale government deficits, reductions in fiscal spending, fixed and sustained reductions in government debt. Mountford and Uhlig [31] indicated the impact of changes in fiscal expenditure on stock price volatility by limiting the impulse response function. The results show that increasing fiscal expenditure will negatively impact stock price volatility. Sy [32] elaborated that government revenue can positively affect stock prices in the Philippine market, and government spending can negatively impact the stock. According to Mbanga and Darrat [33], fiscal policy has long-term and short-term effects on the U.S. stock market, but the long-term impact is more significant. Chatziantoniou et al. [17] used a structural VAR model to find that fiscal and monetary policies must work together to significantly affect stock markets in Germany, the United States, and the United Kingdom. Prukumpai and Sethapramote [34] showed that fiscal policy shocks in Thailand will only affect the stock market in the next two to three quarters. Lawal et al. [35] pointed out that the interaction of fiscal policy and monetary policy will have an impact on the Nigerian stock market. Emamian and Mazlan [36] found that U.S. fiscal policy could have a significant impact on stocks in the short term. Chen [37] confirmed that the indirect relationship between fiscal policy and stock market performance is explained through the money supply channel.

Jiang [38] evidenced that Guangdong Province’s fiscal expenditure plays a significant role in stimulating economic growth. Jia and Zhu [39] found in the analysis of Anhui Province that fiscal expenditure has a positive pulling effect on economic growth and the magnitude of the pulling effect is closely related to the strength of fiscal policy. Nijkamp and Poot [13] pointed out that the impact of fiscal spending on the economy will increase over time. Wu et al. [40] explained that during the COVID-19 epidemic, the Chinese government’s aggressive fiscal expenditure has facilitated economic recovery. Zhu et al. [39] indicated that fiscal expenditure will not promote economic growth when a relatively good financial environment. Still, when the financial climate deteriorates, the increase in fiscal expenditure will promote economic growth. Economic growth positively impacts stock prices [25,26,41], therefore, fiscal spending may increase stock prices by fostering economic growth.

To sum up, in previous studies, the interaction mechanism between pharmaceutical fiscal expenditure and the stock price of the pharmaceutical industry has not been explained from the perspective of China, nor has it been explained whether the financial expenditure of the pharmaceutical industry has an impact on the stock price. Existing studies ignore that the interaction between fiscal spending and stock price may vary over time, making Granger causality’s results non-constant. Therefore, the bootstrap sub-sample rolling-window causality test can be employed to re-examine the correlation between *FE* and *SP*. This paper aims to explore the interaction of *FE* and SP, demonstrate whether the government’s fiscal expenditure on health care can affect the pharmaceutical industry’s development, and provide market investors with a new forecasting tool.

## 3. Theoretical Basis

Ndikumana [42] pointed out that the functions of fiscal expenditure mainly include the theory of intertemporal investment behaviour, the aggregate demand effect of fiscal expenditure, and the positive externality of public investment. According to Qi [12], the role of fiscal expenditure is mainly played in the following ways: (1) Direct support for important resources to improve the quality of economic growth, such as a direct increase in education expenditure and medical expenditure. (2) The government purchases emerging industries or products, optimises the industrial structure and invests more financial and social capital into these potential industries. (3) Special fiscal funds are concentrated in key areas, and credit capital, social capital, and land resources are guided toward specific regions, specific industries, and specific fields. (4) Increased service-related fiscal expenditure, improved residents’ welfare, and guided the whole society to increase capital. Moreover, there are the following classical economic theories to analyse the impact of fiscal policy on stocks.

### 3.1. Theory 1: Keynesian Theory

Keynesians maintain that increased government spending during difficult economic times increases aggregate demand, which in turn may push up stock prices. Blanchard [43] as a proponent of Keynesianism, believes that active fiscal policy can directly expand aggregate demand, stimulate economic development, increase residents’ income and employment levels, and have a great impact on macro-economic activities, which can have an effect on the stock market, increasing stock prices and yields.

### 3.2. Theory 2: Classical Economic Theory

The classical theory considers potential crowding-out effects due to increased government demand for loanable funds and reduced availability of funds for productive sectors such as stock markets [17]. The expansionary fiscal policy has a certain crowding-out effect, which will reduce the loanable funds in the economic field and the investment in the production field, thereby reducing investment by crowding out private sector activities and lowering the stock market price.

### 3.3. Theory 3: Ricardian Theory

Ricardian’s view suggests that policy cannot have an effect on aggregate demand, since any public borrowing would be offset by private saving by rational households. Barro [44] further maintained and develops the Ricardian equivalence proposition, stating that fiscal policy is considered ineffective and does not affect stock markets. Fiscal expenditures will not increase consumer demand, nor will there be a crowding-out effect of consumer spending.

### 3.4. Theory 4: Tobin’s Theory

Tobin [45], in order to study the influence of fiscal policy and monetary policy on stock market price fluctuations, established a model to analyse the transmission mechanism and found that there are two main ways that fiscal policy affects stock price volatility: one is to affect the stock prices through interest rates, and the other is to affect the volatility of stock prices through confidence in good future expectations.

Medical and health fiscal spending influences the stock prices of the pharmaceutical business in the following ways. First, the government purchases pharmaceuticals and medical supplies directly from pharmaceutical corporations. The government is the greatest purchaser of these items, and its purchases may have a significant influence on the stock values of the firms from whom it purchases. The second method is through industry regulation. The government supervises the prices and marketing of pharmaceuticals. Depending on the regulation, the impact on stock prices might be beneficial or negative. Through research and development, a tax credit is the third method. Companies that spend on research and development receive this credit. This can have a favourable influence on stock prices since it motivates firms to invest in new items.

On the other hand, medical and health fiscal expenditures can impact the stock price of the pharmaceutical business by influencing the demand for pharmaceutical products. If medical and health expenditures grow, this might contribute to a rise in the demand for pharmaceutical items and the price of stocks. Another method is through influencing the production costs of pharmaceutical firms. If medical and health expenditures rise, this can result in higher expenses for pharmaceutical businesses, which can lead to a decline in stock values. The pharmaceutical sector relies heavily on medical and health expenditures. Medical and health expenditures consist of all government expenditures on health care, including public health, hospitals, clinics, and prescription pharmaceuticals. Government funding drives the pharmaceutical industry’s demand for their goods. When government spending on health care rises, so does the demand for pharmaceutical products, resulting in a rise in the share prices of pharmaceutical companies.

## 4. Methodology

### 4.1. Bootstrap Full-Sample Causality Test

Since the traditional Granger causality test will distort the test results, we apply the Granger causality test under the VAR framework. The residual-based bootstrap (RB) method is applied to asymptotic distribution, which was proposed by Shukur and Mantalos [46] to avoid the variable instability problem under the VAR framework. The correlation between *FE* and *SP* will be tested by the *RB*-based modified-*LR* statistic.

We applied an RB-based modified-*LR* causality test to construct the bivariate VAR (*n*) process as follows: (1)$$X_{l}=\partial_{0}+\partial_{1}X_{l-1}+\ldots +\partial_{n}X_{l-n}+\delta_{l}\;\;\;l=1,2,\ldots ,L$$
where $\delta_{l}=(\delta_{1l},\delta_{2l})\prime$ means follow zero mean and independent. The optimal lag order *n* is acquired from the Schwarz Information Criterion (SIC). By dividing *X* into *FE* and SP, that is $X_{l}=(FE_{l},SP_{l})\prime$. Since monetary policy and fiscal policy jointly affect the stock market [15,16], we use the money supply (MS) as a control variable. We can write Equation (2) as follows:(2)$$\left[\begin{array}{c} FE_{l}\\ SP_{l} \end{array}\right]=\left[\begin{array}{c} \partial_{10}\\ \partial_{20} \end{array}\right]+\left[\begin{array}{ccc} \partial_{11}(I) & \partial_{12}(I) & \partial_{13}(I)\\ \partial_{21}(I) & \partial_{22}(I) & \partial_{23}(I) \end{array}\right]\left[\begin{array}{c} FE_{l}\\ SP_{l}\\ MS_{l} \end{array}\right]+\left[\begin{array}{c} \delta_{1l}\\ \delta_{2l} \end{array}\right]$$
where $\partial_{ab}(I)=\sum_{k=1}^{n}\beta_{ab,k}I^{k}$, *a*, *b =* 1, 2, 3 and $I^{k}X_{l}=X_{l-k}$. (*I* represents the lag operator.)

Based on Equation (2), if *SP* has an important impact on the *FE*, the null hypothesis should be rejected that *SP* does not Granger cause *FE* ($\partial_{12,k}=0$ for *k* = 1, 2, …, *n*). Likewise, *FE* cannot affect *SP* ($\partial_{21,k}=0$ for *k* = 1, 2, …, *n*) can be examined similarly.

### 4.2. Parameter Stability Test

We address the problem of structural variation in parameters with the *Sup-F*, *Ave-F*, and *Exp-F* tests [47]. Nyblom [48] proposed that whether the parameters follow a random walk process can be verified by the $L_{c}$ statistic test. If the parameters are not constant, we will employ the bootstrap sub-sample rolling-window test to investigate the correlation between *FE* and *SP*.

### 4.3. Bootstrap Sub-Sample Rolling-Window Causality Test

According to Balcilar et al. [49] and Su et al. [50], we set the rolling window width to *I*, and then divide the entire time series into multiple sub-samples to obtain *I* observations. Assuming that *L* is the length of the time series and the sub-samples of each segment end up as *i*, *i + 1*, …, we can obtain *L* and *L – I + 1* sub-samples. Then, based on the *RB*-based modified-*LR* test, we can obtain the Granger causality between *FE* and *SP*. Finally, after calculating the *p*-value and *LR* statistic, the results of the bootstrap sub-sample rolling-window test can be obtained.

The mean of a significant number of estimates $M_{c}^{-1}\sum_{k=1}^{n}{\hat{\partial }}_{12,k}^{*}$ represent the influence from *SP* to *FE* and $M_{c}^{-1}\sum_{k=1}^{n}{\hat{\partial }}_{21,k}^{*}$ represent the influence from *SP* to *FE*. According to Su et al. [51], we chose the 90% confidence interval and corresponding lower (5th quantiles of ${\hat{\beta }}_{12,k}^{*}$) and upper (95th quantiles of ${\hat{\beta }}_{21,k}^{*}$) bounds. 

## 5. Data

This article investigates the correlation between *FE* and *SP* by considering monthly data from 2007:M1 to 2021:M12. In 2007, the scope of fiscal expenditure items had changed a lot, and the data of previous years cannot be compared, so we chose the monthly data starting from 2007. From the National Bureau of Statistics of China, we obtained data on medical and health fiscal expenditures. Then, we used Shanghai Stock Exchange Medical and Health Industry Index to represent the trend of stocks in the pharmaceutical industry. We transformed all raw data into natural logarithms to avoid potential heteroskedasticity and dimensional differences between series. The data are first-differenced to perform a bootstrap causality test. This period is a critical period for the rapid expansion of China’s pharmaceutical industry. It also covers important events such as China’s medical reform and drug reform, which is representative. In 2007, the investment in medical and health financial expenditure was CNY 198.996 billion, which will increase nearly ten-fold by 2019, reaching CNY 1.9205 billion. Jin [8] indicated that China’s public health expenditure is still relatively low compared to developed countries. In other words, China’s medical and health investment still needs to be expanded. As an industry with development potential, the pharmaceutical index is also rising. This suggests that fiscal spending may boost stocks by boosting economic growth, consistent with Keynesian conclusions. The “Wenchuan Earthquake” in 2008, as a significant public health emergency, increased financial expenditure on health care. In the same year, the emergence of the financial crisis caused the stock market to fall as a whole. Therefore, *FE* and *SP* do not always move in the same direction.

As can be seen from Figure 1, the monthly medical and health fiscal expenditures are not equal, and the fiscal expenditures in the second half of the year are more than those in the first half of the year. This shows that there may be problems in China’s medical and health expenditure structure. However, the overall financial expenditure on medical and health care is rising, indicating that the government is paying more and more attention to the medical and health industry. In 2009, China started the particular reform of medical and health care; then, financial health expenditure showed an upward trend. After 2014, China’s economy entered a new economic normal state, and the overall growth rate of fiscal health expenditure has slowed down. During this period, pharmaceutical stocks have generally risen. However, in 2018, the U.S.–China trade war continued to escalate, resulting in rising domestic economic uncertainty [52], heavy losses in the stock market, and a sharp drop in the pharmaceutical index. In 2020, due to the impact of COVID-19, the demand for medical resources increased rapidly, and the pharmaceutical index began to show a continuous upward trend. Therefore, it can be seen that the correlation between *FE* and *SP* is complex and changes over time.

From Table 1, the average values of *FE* and *SP* are 874.995 and 5013.098, respectively. The skewness of *FE* is 0.974, and the skewness of *SP* is 0.630, indicating that both data sets have asymmetry. The kurtosis of *FE* was 3.480, representing a fat-tailed feature, while the kurtosis of *SP* is 2.743, showing a flat distribution. On the side, the results of the test of Jarque–Bera show that the two series are non-normally distributed at the 1% level. Hence, the traditional Granger causality test is not applicable. In this article, the bootstrap sub-sample rolling-window method and the *RB* method are used to increase the reliability of the results and avoid the possible non-normal distribution between *FE* and *SP*.

## 6. Empirical Results

The bivariate VAR model based on *FE* and *SP* is employed to estimate the causal correlation of the full sample according to Equation (2). According to Schwarz Information Criterion (SIC), we find the optimal lag length is five. Table 2 represents the results of the full-sample causality test. The bootstrap *p*-value shows that the correlation between *FE* and *SP* is not obvious, reflecting that *FE* does not lead to SP, and similarly, *SP* does not lead to *FE*.

We can use the *Sup-F*, *Ave-F* and *Exp-F* tests to test whether the parameters of the VAR model are stable. Furthermore, the reliability of the causality test can be improved by the *L_c_* statistical test [53]. It can be seen from Table 3 that the *Sup-F test* highlights abrupt structural changes in *FE* at a 1% level. According to the *Ave-F test*, it can be concluded that the *FE* has structural changes at the 1% level, and there are structural changes at the 10% level in the VAR system. The *Exp-F test* shows that the parameters of the *FE* variable change gradually over time. Thereby, the results obtained by the full-sample causality test are unreliable. We investigated the time-varying relationship between *FE* and *SP* by employing a bootstrap sub-sample rolling-window causality test. In addition, to ensure the accuracy of causal analysis, the rolling-window width is selected as 24 months in this paper. 

According to Figure 2, it is can be obtained the direction and extent of the impact of *FE* on *SP*. The *FE* Granger causes *SP* at the 10% level of significance in the periods of 2009:M3–2009:M7, 2013:M8–2014:M3, 2014:M5–2014M8, 2020:M10–2021:M1 (*FE* has a positive influence on *SP*), and 2009:M7–2010:M5 (*FE* has negative impacts on *SP*). 

The State Council published a guideline on deepening the reform of the medical and health care system in March 2009. The new medical reform policy shows that the government attaches great importance to the reform policy, which has promoted a massive increase in government medical and health expenditures in 2009. The increase in the state’s financial investment in medical and health care has boosted investors’ confidence in investment [45]. Consumer confidence also affects spending [52], and changes in investor sentiment could affect investment choices. So, investor demand for pharmaceutical stocks rises, and share prices will rise. Hence, during the periods of 2009:M3–2009:M7, *FE* positively affected *SP*. However, according to classic theory, after fiscal expenditure increases, there may be a crowding-out effect, where higher interest rates reduce the value of stock prices. Fiscal spending can have a huge and rapid impact on economic activity in the short term, but excessive fiscal spending can lead to liquidity traps, which in turn hinder economic development [15]. Moreover, in times of crisis, such as the global financial crisis, the subprime mortgage crisis, and the spread of the COVID-19 epidemic, fiscal expansion has a greater impact on the stock market. After the economy recovers, the impact of fiscal policy gradually weakens [32]. Moreover, the cost of importing and exporting pharmaceutical products may also have an impact on the stock market. Hence, *FE* negatively affected *SP* during the period of 2009:M7–2010:M5 after the medical reform was carried out.

In 2013, the state’s financial investment in health care increased, and favourable policies will still boost the stock market. Moreover, the acceleration of urbanisation and ageing in China has also promoted the increase in medical and health expenditures [47], and the public’s demand for medical services, drugs, medical equipment and other pharmaceutical products has increased, which has also promoted the development of the pharmaceutical industry. Hence, during 2013:M8–2014:M3, *FE* positively affected *SP*. China’s fiscal system underwent significant changes in 2014 with the passage of a new budget law. Hao et al. [5] illustrated that public finance and its reforms have an obvious positive effect on health spending in China. Chatziantoniou et al. [17] found that fiscal expenditure can indirectly affect the stock market through the interest rate channel. Fiscal policy shocks tend to have only short-term effects, hence, *FE* has a positive impact on *SP*. The spread of COVID-19 has disrupted the global financial markets and adversely affected the economic outlook since mid-February 2020 [54]. In response to the crisis, governments worldwide have implemented economic stimulus programs and increased fiscal spending by directly funding health care systems [27]. Kotlebova [14] showed that fiscal policy can positively support the real economy and affect stock market development. Wang et al. [7] evidenced that increasing fiscal expenditure will not promote economic growth when the financial environment is relatively reasonable but can significantly promote economic growth when the financial environment is deteriorating. Levine [26] studied the relationship between the stock market and the economy, pointing out that stocks would rise with economic growth. Therefore, the positive influence of *FE* on *SP* can be evidenced during the period of 2020:M10–2021:M1. In a word, the positive effect of *FE* on *SP* is consistent with the Keynesian theory, while the negative effect supports the classicist theory.

The direction of the interaction of *SP* on *FE* and *p*-values can be obtained from Figure 3. The alternative hypothesis that *SP* does Granger cause *FE* can be accepted in 2016:M6–2016:M10 (SP has a positive influence on FE), 2019:M3–2019M7, 2019:M10–2020:M7, 2021:M01–2021:M07 (SP has a negative influence on FE) at a 10% significance level.

The “Pilot Program for Drug Marketing Authorization Holder System” was issued by the State Council on 6 June 2016, separating drug registration from production licenses. The impact of specific categories of policies on stock market volatility has fluctuated significantly over time [55]. Cheng et al. [56] showed that the release of pharmaceutical policies is strongly linked to the fluctuation of the pharmaceutical index, and the scheme has promoted the rise of the pharmaceutical index. An increase in the stock market means that the economy is developing and the country is prosperous. As a country gets richer, so does health care spending [8]. Therefore, during the period of 2016:M6–2016:M10, the growth of the pharmaceutical industry index promoted an increase in medical and health expenditure.

However, *SP* also has a negative effect on *FE*. After CCTV exposed the chaos of licensed pharmacists on 15 March 2019, the State Food and Drug Administration launched a particular campaign to strictly investigate the practice of licensed pharmacists in retail enterprises. It announced that licensed pharmacists could practice in more places. In adda is ition, the state has launched a pilot program for centralised drug procurement and carried out “4 + 7 bulk procurement” to reduce drug prices, resulting in the loss of high gross profit in the pharmaceutical industry and the decline in corporate profits leading to a drop in the pharmaceutical index [57].

Meanwhile, medical insurance coverage has increased since 2019, increasing medical insurance fiscal expenditures and reducing health service expenditures. Policy changes 3can cause volatility among stocks [58]. Therefore, pharmaceutical stocks fell while *FE* rose. In August 2019, the revision of the Drug Administration Law contributed to the rise of the pharmaceutical index [59]. The continuous reform of the State Food and Drug Administration has led to a boom in the launch of new drugs, and the pharmaceutical industry has good prospects for development. However, during the COVID-19 outbreak that started at the end of 2019, the supply of pharmaceutical products was in short supply, all industries were shut down, and the national economy was in a downturn [27]. Different countries had different situations and different responses to COVID-19 [4]. The situation in China was relatively serious, so the demand for medical resources increased, and the pharmaceutical sector index rose. However, due to the increasing demand for medical resources, the pharmaceutical sector index rose. Chernick et al. [60] evidenced that many cities face high additional costs and massive income shortfalls during COVID-19, which may lead to budgetary reductions in fiscal expenditures. Therefore, during the 2019:M10–2020:M7 period, an increase in *SP* led to a decrease in *FE*. From January 2021, the spread of COVID-19 in China is under control. The stock price of the pharmaceutical industry has returned to normal levels. However, to recover the economy, China has adopted a series of fiscal policies to stimulate economic development and increased fiscal expenditure [40]. Therefore, from January 2021 to July 2021, *SP* decreased but *FE* increased.

To sum up, under the assumption that the parameter values are fixed, the Grange full-sample test results point out no interaction between *FE* and *S**P*. Furthermore, through stability testing, we realise that there are structural changes in both time series and VAR systems. Therefore, we perform a guided sub-sample rolling-window causality to study the time-varying interaction between two variables. Empirical results show that an increase in *FE* positively impacts *SP*, suggesting that increased fiscal spending on health care does indeed boost the pharmaceutical industry. This result is consistent with Keynes and Tobin’s theory. Furthermore, *FE* will have a negative impact on SP, which supports the classical economic theory. Moreover, *SP* has a positive and negative impact on *FE*, affected by the medical policy and the epidemic.

## 7. Conclusions

We examine the causal correlation between medical and health fiscal expenditure and the pharmaceutical industry stock index. The results of the Grange full-sample test are unreliable, which shows that there is no correlation between the two variables. Thereby, we perform a sub-sample test to study time-varying interactions between two variables. Empirical results show that an increase in *FE* can positively affect SP, suggesting that increased fiscal expenditures on health care promote the pharmaceutical industry’s development. This result supports the Keynesian theory, and Tobin’s theory can explain the conduction path. The negative effect of *FE* on *SP* is consistent with classical theory. The excessive financial investment will have a crowding-out effect and inhibit the development of the pharmaceutical industry. In addition, the impact of the stock index on fiscal expenditure is affected by the period of the economy and the medical and health policies. The government should guide and regulate the development of the pharmaceutical industry by adjusting fiscal expenditure. On the one hand, the government needs to increase the financial expenditure on the pharmaceutical industry to promote the development of the medical and health industry. When the stock market fluctuates violently, the government can maintain the stability of the pharmaceutical market by increasing or decreasing fiscal expenditure. On the other hand, it needs to guide social funds into the pharmaceutical industry, and through the stock market, promote the accumulation of social funds, and then promote the development of the pharmaceutical industry. It is necessary to avoid the “crowding out effect” caused by the expansion of fiscal expenditure, which will reduce private investment and thus affect the operation and development of listed companies. Moreover, the government should introduce reasonable medical and health policies to promote the healthy development of the pharmaceutical industry. Investors can maintain sound investments through fiscal expenditures and related policies and choose optimal asset portfolios. In addition, other developing countries can refer to this study to analyse the correlation between *FE* and *SP* in their own countries, and promote the development of the pharmaceutical industry by regulating financial expenditures.

## Figures and Tables

**Figure 1 ijerph-19-11730-f001:**
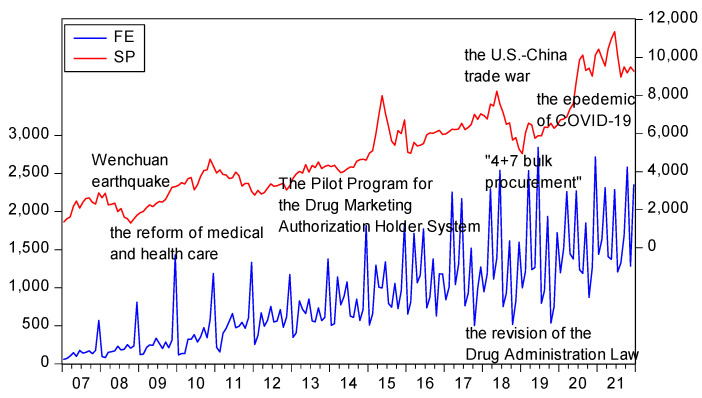
The trends of *FE* and *SP*.

**Figure 2 ijerph-19-11730-f002:**
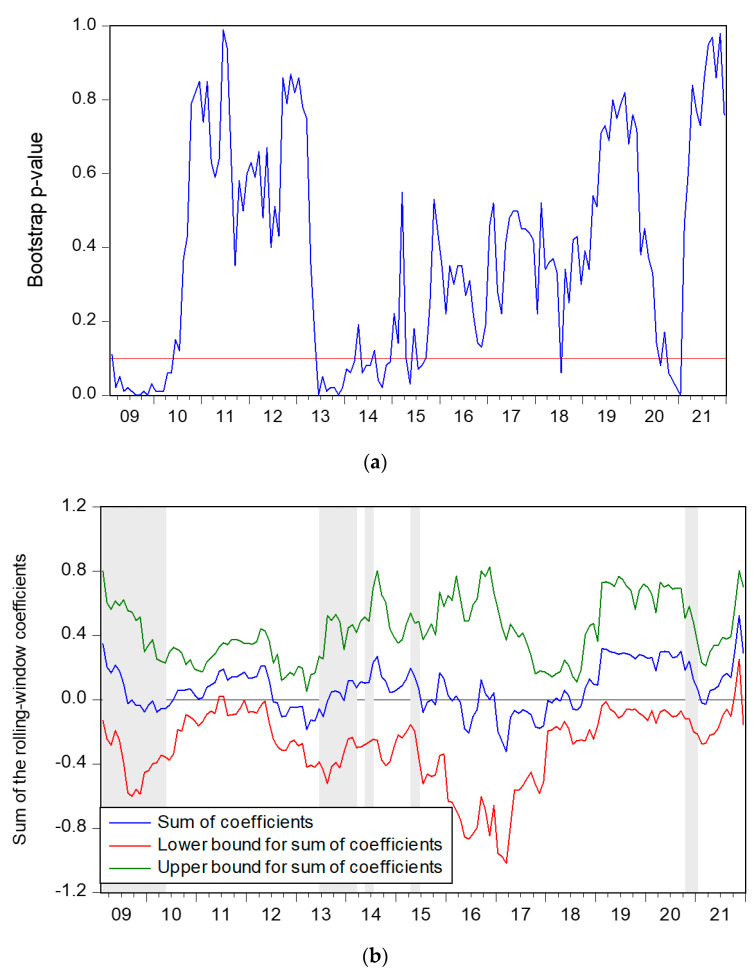
Impact of *FE* on *SP*. (**a**) Bootstrap *p*-values of rolling test statistic testing. (**b**) Impact of *FE* on *SP*.

**Figure 3 ijerph-19-11730-f003:**
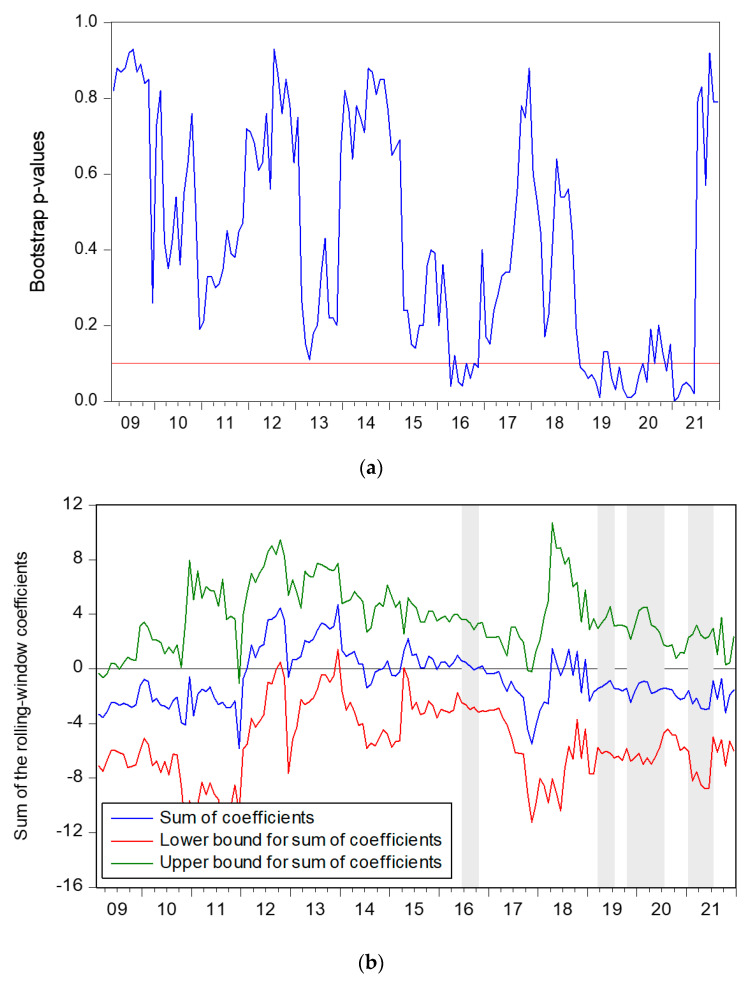
Impact of *SP* on *FE*. (**a**) Bootstrap *p*-values of rolling test statistic testing. (**b**) Impact of *SP* on *FE*.

**Table 1 ijerph-19-11730-t001:** Descriptive statistics for *FE* and *SP*.

	*FE*	*SP*
Mean	874.995	5013.098
Median	739.280	4443.315
Maximum	2842.000	11,360.21
Minimum	58.420	1297.970
Standard Deviation	634.726	2341.506
Skewness	0.974	0.630
Kurtosis	3.480	2.743
Jarque–Bera	30.165 ***	12.419 ***

Note: *** denotes significance at the 1% level.

**Table 2 ijerph-19-11730-t002:** Full-sample Granger causality tests.

Tests	H_0_: *FE* Does Not Granger Cause *SP*		H_0_: *SP* Does Not Granger Cause *FE*	
	Statistics	*p*-Values	Statistics	*p*-Values
Bootstrap *LR* test	1.681	0.870	2.922	0.652

Notes: We calculate *p*-values using 10,000 bootstrap repetitions.

**Table 3 ijerph-19-11730-t003:** The results of parameter stability test.

Tests	*FE*		*SP*		VAR System	
	Statistics	*p*-Value	Statistics	*p*-Value	Statistics	*p*-Value
*Sup-F*	89.374 ***	0.000	18.256	0.558	40.981	0.132
*Ave-F*	45.259 ***	0.000	12.883	0.244	27.488 *	0.100
*Exp-F*	4.056 ***	0.000	7.352	0.373	17.081	0.118
*L_c_*					3.862	0.269

Notes: We calculate *p*-values using 10,000 bootstrap repetitions. ***, * denotes significance at the 1%, 10% levels.

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
