# Peer review of "The Impact of Medical and Health Fiscal Expenditures on Pharmaceutical Industry Stock Index in China"

_ijerph, 2022, doi:10.3390/ijerph191811730_

Round 1

Reviewer 1 Report

This article tries to  explore the the cause and effect of  fiscal expenditures and stock index in pharmaceutical industry in china with several different unique statistical analysis on timeline basis, which is interesting . The analysis itself is well described and deeply evaluated. Followings are comments and questions  to be evaporated for publication.

Major comments.

The analysis is basically focused on Chinese pharmaceutical industry and society, I am wondering if author can expand their findings to other countries or industries or only limited in China. It is needed to add some other references or analysis to address this point.

After the section of 6, from the line 318, and 379, author try to pick up some cases of incidents which fit and support their findings, it would be better to pick up some anomalies to tell their analysis is more objective. 

It would be interesting to argue whether their analysis is only for retrospective or can be applied for perspective/predictive approach.

Minor comments.

line 56 my countries would be  China

line 67 the term, mechanism of action is not appropriate, should be changed.

line 99  between the two, rephrase the sentense.

line 277, 278, 280, 416 and other, the SE would be SP.

line 409, 410, the conclusion ( the effect btw FE and SP is complicated and time-varying) is quite boring, I would suggest to delete this sentence  or rephrase.

Author Response

Response Sheet

Reply to Review 1

Comments to the Author

Q1: Major Comments

The analysis is basically focused on Chinese pharmaceutical industry and society, I am wondering if author can expand their findings to other countries or industries or only limited in China. It is needed to add some other references or analysis to address this point.

Response: Thank you so much for your comments.

We have added relevant content for other countries, as follows (Page 2-3):

The stabilizer role of fiscal expenditure and the mechanism of discretionary power can promote the stability of social resources, guide the flow of fiscal funds to public areas such as education, medical care, and environmental protection, and drive economic development [12]. Nijkamp and Poot [13] clarify that increased fiscal spending is beneficial to future economic development. Kotlebova et al. [14] illustrate that fiscal policy can boost the real economy and positively impact the stock market. However, according to Foresti and Napolitano [15], macroeconomic scenarios can also affect stock market responses to fiscal policy. During the financial crisis, the stock index will rise with the increase of fiscal investment, but in other periods, the loose fiscal policy will cause the stock to plummet [15]. For other purposes, such as to achieve environmental goals, the government will adopt aggressive fiscal subsidy policies, which can also have an impact on an industry [14]. Afonso and Sousa [16] investigate that government expenditures negatively affect stocks market in the U.S., U.K. and Germany. Chatziantoniou et al.[17] show that aggressive government spending sends U.K. stocks lower, but fiscal spending indirectly affects stocks in Germany and the U.S.. Jakova [18] illustrates that in Slovakia, Romania, the Czech Republic and Bulgaria, rising stocks lead to lower government spending, in Poland there is no relationship between stock market performance and government spending, and Hungary's stock market performance was positively affected by higher public spending. Therefore, the relationship between fiscal spending and the stock market is different in different countries. Ardagna [14] finds that the trend of a country’s past fiscal policy and its position in its policy and fiscal policy changes can lead to sharp declines in company stock prices. According to Afonso and Fernandes [15], the changes in financial expenditure will negatively affect the stock price of listed companies.  ……

Q2: Major Comments

After the section of 6, from the line 318, and 379, author try to pick up some cases of incidents which fit and support their findings, it would be better to pick up some anomalies to tell their analysis is more objective. 

It would be interesting to argue whether their analysis is only for retrospective or can be applied for perspective/predictive approach.

Response: Thank you so much for your comments.

We have cited other cases to analyze the correlation between health care spending and the pharmaceutical index during this period, as follows (Page 10):

……However, according to classic theory, after fiscal expenditure increases, there may be a crowding-out effect, where higher interest rates reduce the value of stock prices. Fiscal spending can have a huge and rapid impact on economic activity in the short term, but excessive fiscal spending can lead to liquidity traps, which in turn hinder economic development [15]. Moreover, in times of crisis, such as the global financial crisis, the subprime mortgage crisis, and the spread of the COVID-19 epidemic, fiscal expansion has a greater impact on the stock market. After the economy recovers, the impact of fiscal policy gradually weakens[33]. Besides, the cost of importing and exporting pharmaceutical products may also have an impact on the stock market. Hence, FE negatively affected SP during the period of 2009:M7-2010:M5 after the medical reform was carried out.

Q3: Minor Comments

line 56 my countries would be  China

line 67 the term, mechanism of action is not appropriate, should be changed.

line 99  between the two, rephrase the sentense.

line 277, 278, 280, 416 and other, the SE would be SP.

line 409, 410, the conclusion ( the effect btw FE and SP is complicated and time-varying) is quite boring, I would suggest to delete this sentence or rephrase.

Response: Thank you so much for your comments.

We have made revisions to the problems raised above. For details, please refer to the places marked in red in the article. Since the revisions are scattered and short, we will not repeat them here.

Reviewer 2 Report

This article is very well written and constructed.  I congratulate the authors on the quality of the paper. 

Author Response

Response Sheet

Reply to Review 2

Comments to the Author

This article is very well written and constructed.  I congratulate the authors on the quality of the paper. 

Response: Thank you so much for your comments.

We will work harder on this topic in the future.

Reviewer 3 Report

This paper uses sophisticated econometric methods to investigate the relationship, if any, between fiscal spending on health care in China and the Chinese stock index for pharmaceuticals.

Unfortunately, this paper has major flaws. The theoretical motivation is extremely thin. The authors appeal to macro theories (which are described extremely superficially) about the general effects of fiscal policy on economic growth and of fiscal policy and economic growth on equity prices. However, they provide no reason why one would expect any link between a budget component such as health care, and a sectoral stock index, such a pharmaceuticals. Nor is any such link known in the literature. This simply means that the motivating theory is ill defined. Given this, the ambiguity in the results are not surprising. This makes the apparent significance found from splitting the sample into subsamples look spurious.

Even if the theoretical link were well defined, the absence of control variables is glaring. The overall fiscal stance would have been an obvious choice, as would the development of the overall stock market. The authors mention some important policy changes that could have been relevant. Population ageing, which they also mentioned, is another candidate.

Furthermore, China is studied as if it were isolated from the rest of the world. In reality, pharmaceuticals are widely traded internationally. Chinese health care can use foreign-made products, and Chinese pharmaceuticals can produce for exports. This is another reason why the link between Chinese health care spending and Chinese pharmaceutical stock indices is ill defined.

In conclusion, sophisticated econometric methods is no substitute for clear theoretical reasoning.

Author Response

Response Sheet

Reply to Review 3

Comments to the Author

Q1:

This paper uses sophisticated econometric methods to investigate the relationship, if any, between fiscal spending on health care in China and the Chinese stock index for pharmaceuticals.

Unfortunately, this paper has major flaws. The theoretical motivation is extremely thin. The authors appeal to macro theories (which are described extremely superficially) about the general effects of fiscal policy on economic growth and of fiscal policy and economic growth on equity prices. However, they provide no reason why one would expect any link between a budget component such as health care, and a sectoral stock index, such a pharmaceuticals. Nor is any such link known in the literature. This simply means that the motivating theory is ill defined. Given this, the ambiguity in the results are not surprising. This makes the apparent significance found from splitting the sample into subsamples look spurious.

Response: Thank you so much for your comments.

We have reorganized and supplemented the theoretical motivations and further added to the reasons why one would expect any link between health care spending and the pharmaceutical stock index, as follows (Pages 1-3, Page 4) :

Page 1-3:

…… According to Heidari et al.[1] , the pharmaceutical industry is considered to be the most competitive and profitable key industry in the world, and its stock index is favored by the majority of investors. Hence, investors prefer to gain benefits by understanding the factors affecting the pharmaceutical industry. It is necessary to pay attention to the ways in which government policies affect the pharmaceutical industry. Furthermore, the “healthy China” strategy put forward at the 19th National Congress of the Communist Party of China (CPC) and the policy for ensuring essential people’s livelihood in 2020 demonstrate that the government attaches great importance to people’s health. Therefore, the pharmaceutical industry in China has good development prospects and has attracted much attention from investors.

……

The securities market of China has gradually developed over the past two decades. It has been a significant industry in the national economy, extensively promoting the economy’s growth. However, the low entry threshold for companies to go public and the low overall investment quality of traders, mostly retail investors, lead to serious speculation in my country’s stock market. Hence, government intervention in the market is inevitable[7]. Moreover, China's stock market is immature, and stock prices are prone to ups and downs. The factors affecting stock prices are more complicated than those in developed countries. Therefore, studying the impact of government investment in health care on the stock index of the pharmaceutical industry will help investors to clarify the investment direction. Jin and Qian [8] point out that China’s health care expenditure level is relatively low compared with developed countries. The improvement of economic status will promote increasing spending on health care [9,10]. Since the new medical reform in 2009, the scale of Chinese health care expenditure has gradually expanded. From 198.9 billion yuan in health expenditure in 2007 to 20,894 yuan in 2021, China’s medical and health expenditure has increased by about 10.5 times, hence, the increase in public health spending may be a key factor in promoting the development of the pharmaceutical industry.  ……

The stabilizer role of fiscal expenditure and the mechanism of discretionary power can promote the stability of social resources, guide the flow of fiscal funds to public areas such as education, medical care, and environmental protection, and drive economic development [12]. Nijkamp and Poot [13] clarify that increased fiscal spending is beneficial to future economic development. Kotlebova et al. [14] illustrate that fiscal policy can boost the real economy and positively impact the stock market. However, according to Foresti and Napolitano [15], macroeconomic scenarios can also affect stock market responses to fiscal policy. During the financial crisis, the stock index will rise with the increase of fiscal investment, but in other periods, the loose fiscal policy will cause the stock to plummet [15]. For other purposes, such as to achieve environmental goals, the government will adopt aggressive fiscal subsidy policies, which can also have an impact on an industry [14]. Afonso and Sousa [16] investigate that government expenditures negatively affect stocks market in the U.S., U.K. and Germany. Chatziantoniou et al.[17] show that aggressive government spending sends U.K. stocks lower, but fiscal spending indirectly affects stocks in Germany and the U.S.. Jakova [18] illustrates that in Slovakia, Romania, the Czech Republic and Bulgaria, rising stocks lead to lower government spending, in Poland there is no relationship between stock market performance and government spending, and Hungary's stock market performance was positively affected by higher public spending. Therefore, the relationship between fiscal spending and the stock market is different in different countries. Ardagna [14] finds that the trend of a country’s past fiscal policy and its position in its policy and fiscal policy changes can lead to sharp declines in company stock prices. According to Afonso and Fernandes [15], the changes in financial expenditure will negatively affect the stock price of listed companies.

……

Page 4:

  1. Theoretical Basis

Ndikumana [43] points out that the functions of fiscal expenditure mainly include the theory of intertemporal investment behavior, the aggregate demand effect of fiscal expenditure, and the positive externality of public investment. According to Qi [12], the role of fiscal expenditure is mainly played in the following ways: (1) Direct support for important resources to improve the quality of economic growth, such as direct increase in education expenditure and medical expenditure. (2) The government purchases emerging industries or products, optimizes the industrial structure, and invests more financial and social capital into these potential industries. (3) Special fiscal funds are concentrated in key areas, and credit capital, social capital, and land resources are guided toward specific regions, specific industries, and specific fields. (4) Increase service-related fiscal expenditure, improve residents' welfare, and guide the whole society to increase capital. Besides, there are the following classical economic theories to analyze the impact of fiscal policy on stocks.

……

Q2:

Even if the theoretical link were well defined, the absence of control variables is glaring. The overall fiscal stance would have been an obvious choice, as would the development of the overall stock market. The authors mention some important policy changes that could have been relevant. Population ageing, which they also mentioned, is another candidate.

Response: Thank you so much for your comments.

We have added control variables to the article as follows (Page 5-6):

……

where  means follow zero mean and independent. The optimal lag order n is acquired from the Schwarz Information Criterion (SIC). By dividing X into FE and SP, that is. Since monetary policy and fiscal policy jointly affect the stock market [15,16], we use the money supply (MS) as a control variable.  We can write Equation (2) as follows:

                                                (2)

where , a, b=1, 2, 3 and . (I represents the lag operator.)

……

Q3:

Furthermore, China is studied as if it were isolated from the rest of the world. In reality, pharmaceuticals are widely traded internationally. Chinese health care can use foreign-made products, and Chinese pharmaceuticals can produce for exports. This is another reason why the link between Chinese health care spending and Chinese pharmaceutical stock indices is ill defined.

In conclusion, sophisticated econometric methods is no substitute for clear theoretical reasoning.

Response: Thank you so much for your comments.

We have supplemented relevant information from other countries and refined the explanation of the economic situation in the conclusion section, as follows (Page 2-3, Page 10,) :

Page 2-3:

……However, according to Foresti and Napolitano [15], macroeconomic scenarios can also affect stock market responses to fiscal policy. During the financial crisis, the stock index will rise with the increase of fiscal investment, but in other periods, the loose fiscal policy will cause the stock to plummet [15]. For other purposes, such as to achieve environmental goals, the government will adopt aggressive fiscal subsidy policies, which can also have an impact on an industry [14]. Afonso and Sousa [16] investigate that government expenditures negatively affect stocks market in the U.S., U.K. and Germany. Chatziantoniou et al.[17] show that aggressive government spending sends U.K. stocks lower, but fiscal spending indirectly affects stocks in Germany and the U.S.. Jakova [18] illustrates that in Slovakia, Romania, the Czech Republic and Bulgaria, rising stocks lead to lower government spending, in Poland there is no relationship between stock market performance and government spending, and Hungary's stock market performance was positively affected by higher public spending. Therefore, the relationship between fiscal spending and the stock market is different in different countries. Ardagna [14] finds that the trend of a country’s past fiscal policy and its position in its policy and fiscal policy changes can lead to sharp declines in company stock prices. According to Afonso and Fernandes [15], the changes in financial expenditure will negatively affect the stock price of listed companies.

Page 10:

……Fiscal spending can have a huge and rapid impact on economic activity in the short term, but excessive fiscal spending can lead to liquidity traps, which in turn hinder economic development [15]. Moreover, in times of crisis, such as the global financial crisis, the subprime mortgage crisis, and the spread of the COVID-19 epidemic, fiscal expansion has a greater impact on the stock market. After the economy recovers, the impact of fiscal policy gradually weakens[33]. Besides, the cost of importing and exporting pharmaceutical products may also have an impact on the stock market. Hence, FE negatively affected SP during the period of 2009:M7-2010:M5 after the medical reform was carried out.

Round 2

Reviewer 3 Report

The revision does not address my main objection to this study, namely, that it investigates the correlations and possible causation between two quantities that should not be expected to show any such systematic ties. That renders the results meaningless.

Author Response

Reply to Reviewer 3

The revision does not address my main objection to this study, namely, that it investigates the correlations and possible causation between two quantities that should not be expected to show any such systematic ties. That renders the results meaningless.

Response: Thank you very much for you comments. The systematic connections between these two have been elaborated on in the theoretical portions.

The medical and health fiscal spending influences the stock prices of the pharmaceutical business in the following ways. First, the government purchases pharmaceuticals and medical supplies directly from pharmaceutical corporations. The government is the greatest purchaser of these items, and its purchases may have a significant influence on the stock values of the firms from whom it purchases. The second method is through industry regulation. The government supervises the prices and marketing of pharmaceuticals. Depending on the regulation, the impact on stock prices might be beneficial or negative. Through the research and development tax credit is the third method. Companies who spend in research and development receive this credit. This can have a favorable influence on stock prices since it motivates firms to invest in new items.

On the other hand, medical and health fiscal expenditures can impact the stock price of the pharmaceutical business by influencing the demand for pharmaceutical products. If medical and health expenditures grow, this might contribute to a rise in the demand for pharmaceutical items and the price of stocks. Another method is through influencing the production costs of pharmaceutical firms. If medical and health expenditures rise, this can result in higher expenses for pharmaceutical businesses, which can lead to a decline in stock values. The pharmaceutical sector relies heavily on medical and health expenditures. Medical and health expenditures consist of all government expenditures on health care, including public health, hospitals, clinics, and prescription pharmaceuticals. Government funding drives the pharmaceutical industry's demand for their goods. When government spending on health care rises, so does the demand for pharmaceutical products, resulting to a rise in the share prices of pharmaceutical companies.
